# Dynamic Models of Within-Herd Transmission and Recommendation for Vaccination Coverage Requirement in the Case of African Swine Fever in Vietnam

**DOI:** 10.3390/vetsci9060292

**Published:** 2022-06-14

**Authors:** Thi Ngan Mai, Satoshi Sekiguchi, Thi My Le Huynh, Thi Bich Phuong Cao, Van Phan Le, Van Hieu Dong, Viet Anh Vu, Anuwat Wiratsudakul

**Affiliations:** 1Faculty of Veterinary Medicine, Vietnam National University of Agriculture, Hanoi 100000, Vietnam; mtngan@vnua.edu.vn (T.N.M.); huynhtmle@vnua.edu.vn (T.M.L.H.); ctbphuong@vnua.edu.vn (T.B.P.C.); letranphan@vnua.edu.vn (V.P.L.); dvhieuvet@vnua.edu.vn (V.H.D.); 2Department of Veterinary Science, Faculty of Agriculture, University of Miyazaki, Miyazaki 889-2192, Japan; sekiguchi@cc.miyazaki-u.ac.jp; 3Center for Animal Disease Control, University of Miyazaki, Miyazaki 889-2192, Japan; 4Faculty of Animal Science, Vietnam National University of Agriculture, Hanoi 100000, Vietnam; vvanh@vnua.edu.vn; 5Department of Clinical Sciences and Public Health, Faculty of Veterinary Science, Mahidol University, Nakhon Pathom 73170, Thailand; 6The Monitoring and Surveillance Center for Zoonotic Diseases in Wildlife and Exotic Animals, Faculty of Veterinary Science, Mahidol University, Nakhon Pathom 73170, Thailand

**Keywords:** ASF, basic reproduction number, dynamic model, herd immunity, vaccination coverage, Vietnam

## Abstract

African swine fever (ASF) is a highly contagious disease that is caused by the ASF virus (ASFV) with a high fatality rate in domestic pigs resulting in a high socio-economic impact. The pig business in Vietnam was recently affected by ASF for the first time. This study thus aimed to develop a disease dynamic model to explain how ASFV spreads in Vietnamese pig populations and suggest a protective vaccine coverage level required to prevent future outbreaks. The outbreak data were collected from ten private small-scale farms within the first wave of ASF outbreaks in Vietnam. Three methods were used to estimate the basic reproduction number (*R*_0_), including the exponential growth method, maximum likelihood method, and attack rate method. The average *R*_0_ values were estimated at 1.49 (95%CI: 1.05–2.21), 1.58 (95%CI: 0.92–2.56), and 1.46 (95%CI: 1.38–1.57), respectively. Based on the worst-case scenario, all pigs in a herd would be infected and removed within 50 days. We suggest vaccinating at least 80% of pigs on each farm once a commercially approved ASF vaccine is available. However, an improvement in biosecurity levels in small-scale farms is still greatly encouraged to prevent the introduction of the virus.

## 1. Introduction

African swine fever (ASF) is a viral hemorrhagic disease caused by the ASF virus (ASFV), a virus belonging to the genus Asfivirus of the family *Asfarviridae* [1]. ASF causes an exceptionally high fatality rate in domestic pigs and wild boars, resulting in a high socio-economic impact. ASF is hence a notifiable disease listed by the World Organization of Animal Health (OIE). The disease is characterized by high fever, loss of appetite, and hemorrhages in the skin and internal organs, with a 30–70% mortality rate [2].

ASF was first reported in Kenya in 1921 [3]. It had then spread rapidly to other African countries. In Europe, ASFV first pervaded Portugal in 1957 before spreading over the Iberian Peninsula. Concurrently, the virus also invaded the American continent with notifications from Brazil, Dominican Republic, Cuba, and Haiti [2]. In 2007, ASFV entered Georgia through the port of Poti, then spread rapidly throughout the country and to its neighboring countries, including Armenia, Azerbaijan, and Russia [4]. From Russia, the virus moved further and invaded the European Union in 2014. In August 2018, the virus first attacked the world’s largest pig producer, China, and it is now spreading in several Asian countries, including Vietnam [5].

In Vietnam, the first ASF outbreak was reported in early February 2019 on a small farm located in Hung Yen province [6]. Subsequently, the virus spread across the country, affecting all provinces (63/63) within seven months [7]. The transmission of ASFV in Vietnam is mainly through the movement of infected pigs, pig products, and infected fomites [7]. Although the virus is transmittable by ticks from the genus *Ornithodoros* [1], there is no evidence of the presence of these ticks in Vietnam.

In this outbreak, the disease was mainly reported on small private farms. It is noticeable that the biosecurity measures were applied on these farms less strictly than in large-scale industrial ones. Meanwhile, the current control measures mainly rely on strict sanitary interventions as neither vaccines nor specific treatments are available [5]. To control the outbreaks, the farm owners were seriously devastated due to the culling process required for complete clearance of the ASF-infected farms, even if there were only a few infected individuals [8]. Currently, there is no commercially available vaccine against ASFV. More recently, a vaccine candidate, ASFV-G-Delta I177L, was developed by deleting the I177L gene from the highly virulent pandemic ASFV strain Georgia (ASFV-G). It showed efficient replicates in a continuous cell line [9]. This opens the door for large-scale vaccine production, a valuable tool for possibly eradicating the virus. With support from the United States Department of Agriculture (USDA), the Vietnamese Department of Animal Health (DAH) and the Navetco National Veterinary Joint Stock Company have now been conducting a vaccination test based on the ASFV-G-Delta I177 L strain [10]. The results were promising, and hopefully, the vaccine will get approval for commercial production shortly.

Mathematical modeling is an effective tool used in studying disease dynamics and provides suggestions on related interventions. One of the most commonly used models for highly contagious diseases is the compartmental model, such as the susceptible-infectious-recovered (SIR) model [11,12]. The model estimates the number of susceptible and infectious individuals in a population and how these numbers change over time [13,14]. In the SIR framework, transmission coefficients quantify how individuals transition from the susceptible to infections and infectious to removed states. The model is frequently used in rapidly transmitted exotic diseases that most of the population lack immunity such as avian influenza [15], lumpy skin disease [16], and ASF [17].

The basic reproduction number (*R*_0_) is one of the basic parameters used in underpinning rational control strategies based on disease modeling. This value can be estimated using a variety of mathematical methods. It quantifies the spread of infectious disease, predicting its speed, scale, and the level of herd immunity required to control the disease. The higher the value of *R*_0_, the faster the epidemic progresses. The *R*_0_ value provides a better understanding of the dynamics of infectious disease outbreaks and the development of effective disease control measures [18]. However, only a few studies have been published quantifying the transmission parameters of ASFV. For example, some were performed under experimental conditions for the between-herd transmission of Georgia 2007/1 strain in wild boars [19,20,21,22]. On-field data from the outbreaks were indicated as a potential source to infer transmission parameters for diseases that cause high mortality in previous studies [23,24]. The present study, therefore, aimed to develop a mathematical model from field data collected by veterinarians to give an insight into how ASF spreads in Vietnamese pig populations, calculate *R*_0_, estimate herd immunity threshold, and ultimately suggest a protective vaccine coverage level that is required to prevent the future outbreaks.

## 2. Materials and Methods

### 2.1. Data Collection

In Vietnam, the farm scale is divided into small and large scales. A farm with less than a thousand pigs is defined as a small-scale farm [25]. These small-scale farms account for over 99% of total pig farms in Vietnam, and they are further subdivided into five subscales: from 1 to 9 pigs, 10 to 29 pigs, 30 to 99 pigs, 100 to 299, and 300 to 999 pigs. Two subscales (100 to 299 and 300 to 999 pigs) accounted for over 52% of the pig farms in this country [25].

In this study, data collected from ten private farms categorized in different scales (seven farms from 100 to 299 and the rest from 300 to 999) were purposively included after these farms were confirmed with ASF-infected status by real-time PCR technique. Mortality and morbidity data of infectious diseases such as ASF are routinely collected on these farms as part of farm management practices and monitoring of dead animals on a farm. These farms were selected based on the availability and quality of the data collected on morbidity. The herds implemented intensive indoor production systems with less strict biosecurity measures compared to large-scale industrial farms. Sows and fattening pigs were found on these farms, with the mean ratio of one sow to ten fattening pigs. These farms were typically classified as farrow-to-finish production types. The number of ASF cases, regardless of whether it was fattening pig or sow, was combined on each farm. We collected data on the starting date of the outbreak, location, farm size, and daily animal morbidity due to ASF. We also created relevant epidemic curves (Appendix A). The starting dates of ASF in these farms were recorded from February to April 2019, within the first wave of ASF outbreaks in Vietnam. Of the recruited farms, seven are located in Hung Yen province, where the first ASF outbreak in Vietnam was reported. The other three farms are located in Ninh Binh and Ha Nam provinces. All provinces in this study are located in northern Vietnam, where the largest populations of pigs are found [26].

### 2.2. Calculation of the Basic Reproduction Number (R_0_)

*R*_0_ is used to express an average number of secondary infections caused by one infected individual in a fully susceptible population during its entire infectious period [18]. The disease will disappear from the population if *R*_0_ < 1 and keep spreading otherwise [27]. The simplicity of the generation time distribution was indicated in a previous study [28]. From the daily case data, we set up the best-fitting generation time (GT) distribution for a series of serial interval (SI) distribution by the “est.GT” function in the package “R0” in program R version 3.4.3 (R development team, Vienna, Austria). In this study, the generation time is the time lag between infection in primary and secondary cases by clinical signs. In addition to GT distribution, all farms’ mean and standard deviation were also calculated. We then estimated *R*_0_ for ASF transmission within pig herds based on our empirical data using three methods, including the exponential growth method (EG), maximum likelihood method (ML), and attack rate method (AR) by the “est.R0.EG”, “est.R0.ML” and “est.R0.AR” functions in the “R0” package of the R program.

### 2.3. Dynamic Modeling of ASF Transmission at Farm Level

To build a mathematical model, different disease transmission parameters are used to demonstrate how the disease spreads in a particular population, such as transmission rate (*β*), removal rate (*γ*), and the basic reproduction number (*R*_0_) [29]. Based on the estimated *R*_0_, the transmission rate (*β*), which represents the daily rate at which infectious animals infect susceptible ones, was calculated with the formula $\beta =R_{0}/T$; where T denotes the infectious period [30]. In this study, we used the mean of the minimum and maximum infectious period reported in a previous study as 3–6 days (4.5 days) and 3–14 days (8.5 days), respectively [19].

With the estimated transmission parameters, the susceptible-infectious-removed (SIR) model of ASF spreading within the herds was constructed using the package “deSolve” in program R. Assuming that the population is entirely susceptible at the beginning of the epidemic, the model is described by the system of differential equations based on time (t):(1)$$\frac{dS}{dt}=\frac{-\beta SI}{N}$$
(2)$$\frac{dI}{dt}=\frac{\beta SI}{N}-\gamma I$$
(3)$$\frac{dR}{dt}=\gamma I$$
where animals are classified as susceptible (S)—animals without clinical signs, infectious (I)—animals with clinical signs, or removed (R)—animals that are removed from the herd either by being culled or died at time t. N is the population size, and Gamma (γ) represents the rate at which individuals are removed from being infectious (equal to 1/infectious period) [30]

To deal with uncertainty, we modeled the within-farm transmission of the ASFV in four different scenarios, as detailed in Table 1. We then initiated the ASFV transmission by introducing one infected pig into the herd on day 1.

### 2.4. Calculation of Herd Immunity

Herd immunity occurs when a significant portion of a herd is immunized enough to provide sufficient protection for those with no immunity [31]. *R*_0_ value is then used to calculate the herd immunity threshold ($I_{c}$), which is the minimum percentage of individuals in the population that would need to be vaccinated to ensure that the disease does not persist. The herd immunity threshold is calculated using the formula  $1-\frac{1}{R_{0}}\;$ [31].

### 2.5. Calculation of Vaccine Coverage

Vaccine coverage ($V_{c}$) is the proportion of the population that must be vaccinated to achieve the herd immunity threshold [32]. The $V_{c}$ value is necessary to halt the spread of ASF in a population. This value depends on the vaccine effectiveness (E) which is the percentage of animals receiving the vaccine and developing immunity. According to a Navetco National Veterinary Joint Stock Company report, under the Ministry of Agriculture and Rural Development (MARD), around 80 percent of vaccinated pigs developed sufficient immunity against ASFV [33,34]. Therefore, we presumed vaccine effectiveness of 80% in this study (E = 0.8). $V_{c}$ is then calculated from $\frac{I_{c}}{E}$ [31,32].

## 3. Results

### 3.1. The Best-Fitting Generation Time Distribution for a Serial Interval of Each Farm

The mean and standard deviation values in days and the best fitting GT distribution for each farm during a 9-week period are shown in Table 2.

### 3.2. The Basic Reproduction Number (R_0_)

The overall *R*_0_ values derived from the three methods and those from each farm are shown in Figure 1A and Figure 1B, respectively. The error bar in each column shows the 95% confidence interval (CI) of the *R*_0_ values.

Overall, the *R*_0_ value ranged from 1.01 to 2.32 with mean values of 1.49 (95%CI: 1.05–2.21), 1.58 (95%CI: 0.92–2.56), and 1.46 (95%CI: 1.38–1.57) based on the EG, ML, and AR methods, respectively (Figure 1A). However, the *R*_0_ values of each farm estimated in different ways had a slight difference (Figure 1B). In farm 3 and farm 6, the *R*_0_ value is relatively high compared to other farms (*R*_0_ > 2). In general, the value of *R*_0_ in each farm estimated by the EG method falls in between those of ML and AR methods. Nonetheless, the 95% CI of the *R*_0_ value calculated with the AR method is lower than other methods. To prepare for the worst-case scenario in which the highest number of infected animals is expected, we thus used the values derived from the ML method with the highest *R*_0_ value in further analysis.

### 3.3. Epidemiological Transmission Parameters

As shown in Table 3, the *R*_0_ value in smaller farms (100 to 299 pigs) is higher than that of the larger ones (300 to 999 pigs). The γ values were 0.22 and 0.12 for the minimum and maximum of the ASFV infectious period, respectively. Based on our calculation, the β values ranged from 0.16 to 0.37 on different farm scales.

### 3.4. Dynamics of ASF Transmission within Farms

In Figure 2, the highest numbers of ASF-infected pigs are observed around days 30 and 60 in the scenarios of the 100–299 farm scale (Figure 2A,B), whereas the time to reach the peak is twice in the farm scale of 300 to 999 (Figure 2C,D), respectively. In the worst-case scenario, all animals are removed within 50 days.

### 3.5. Vaccine Coverage Requirement

The herd immunity thresholds required to establish herd immunity ($I_{c}$) ranged from 28.57% (95%CI: 0.99–47.37%) to 39.75% (95%CI: 0–64.79%) for the 300–999 and 100–299 farm scales, respectively (Table 4). The vaccination coverage required to establish sufficient herd immunity for ASFV ranged from 35.71% (95%CI: 1.24–59.21%) to 49.70% (95%CI: 0–80.99%), depending on farm scales. For small size herds, our results show that approximately 80% of the pigs in each farm should be vaccinated based on the upper bound of the 95% CI of the vaccine coverage (49.70%).

## 4. Discussions

Although, in recent years, there has been a considerable conversation in pig production towards commercial farming; however, Vietnam’s supply of pork still depends mainly on smallholder pig production. In this study, we estimated the basic reproduction number. We used empirical data from ten small-scale farms attacked by the ASFV during the first wave of ASF outbreaks in Vietnam. In addition, we estimated the level of vaccine coverage required to provide enough herd immunity to prevent future ASFV epidemics in Vietnamese pig populations.

This study applied three methods to estimate *R*_0_ from ASF epidemic data. We found that the ML technique gave the highest estimation of the *R*_0_ value. We thus used the values derived from this method for further analysis to prepare our interventions for the worst-case scenario. In addition, the ML method has previously been used in the estimation of *R*_0_ for different infectious diseases such as COVID-19 [35,36,37], Zika virus [38], and influenza [39]. We thus believe that the method would also provide a good estimation in our study.

Our estimated values align with those addressed in previous studies. For example, the values calculated for the Georgia 2007/1 ASFV strain ranged from 1.3 to 4.8 [19]. In addition, a recent outbreak in Uganda revealed *R*_0_ values of 1.6 to 3.2 [21]. A previous study indicated that ASFV circulating strains in Vietnam belong to genotype II [6]. The *R*_0_ values within the pens estimated for the Georgia 2007/1 strain belonging to the same genotype ranged from 1.3 to 4.8 [19]. Compared to these studies, our results fall into the same distribution range.

Nevertheless, some extreme *R*_0_ values were observed under experimental conditions. For instance, the *R*_0_ of isolated Malta 78 strain, genotype I, was reported at 6.9 to 46.9 [40]. Indeed, the *R*_0_ value may vary due to different conditions such as populations, pig species, ASFV genotype, type of data available, risk factors, swine population density, and calculation methods [19,41,42,43]. Therefore, the *R*_0_ values derived from different studies and geographical areas are relatively comparable to a limited extent.

We estimated that the *R*_0_ values ranged from 1.01 to 2.32 in different farm sizes (Figure 1B). Notable, farm 3 and farm 6 had a higher *R*_0_ compared to others (*R*_0_ > 2). The size of these farms, which was relatively small, and lower biosecurity levels may contribute to the larger *R*_0_ values. We also observed from Table 3 that the *R*_0_ in smaller farm scales is higher than those of the larger ones. The *R*_0_ within the farm ranged from 3.9 to 15.6 and was observed in the farm size 47 to 120 pigs in a previous study [43]. Pig density was also a significant risk factor for disease spread [43]. A previous study indicated that the within-herd spread of diseases could be reduced by applying strict biosecurity measures [44]. Lower biosecurity levels of the smaller farms may accelerate the disease spread compared to the larger farms. Therefore, the improvement of preventive measures should be seriously considered to prevent and control the spread of ASFV.

Applying the culling method for the infected farms in Vietnam causes a massive reduction in the pig population. This directly disrupts the overall supply chain. The estimates of the transmission parameters provide quantitative knowledge of the ASF epidemiology, which is helpful for the design and evaluation of more efficient control measures. The transmission rate, β, of newly infected pigs per day depends on the number of infected and susceptible individuals [45]. In our study, the β values were estimated between 0.16 and 0.37 (Table 3), which reflected the number of newly infected pigs per day. A previous study suggested that the β value may vary due to different infectious durations [19]. Besides, modification of γ could be another possibility to include variability, which mathematically is undistinguished, at least for *R*_0_ calculation. We hence built the SIR model for different small farm scales in Vietnam. This compartmental model helped us estimate the disease dynamics of ASF in different scenarios and estimate the number of infected, susceptible, removed, and peak times [11]. The peak in the total number of infections was reached on different days on different farm scales (Figure 2). Our SIR model assesses the impact of decreasing the number of infected-susceptible incidence rates and simulates the within-herd dynamic transmission of the ASFV. Notably, the SIR model estimates the duration that the virus may persist within-herd and how we could lessen the outbreak’s impact [46]. Alternative to the current policy, culling only infected pigs can reduce the number of infected animals in the herd. This was previously addressed as an effective way to control an ASF outbreak [19].

Different vaccines have been developed for ASFV, but none is yet commercially registered. It is thus necessary to envision the protective herd immunity level and vaccine coverage requirement for the upcoming vaccines in the future. Our reviews found that inactivated or subunit ASF vaccines are generally non or partially protective [47]. In contrast, live attenuated vaccines give a promising result at 85.7% to 100% protection depending on the strains [48]. In 2019, Vietnam lost almost 6 million pigs from the ASF outbreaks. Therefore, the Vietnamese government has been trying to develop an attenuated commercial vaccine for ASF from ASFV-G-Delta I 177 L strain in peripheral blood mononuclear cells [10]. A previous study suggested a wide range of the herd immunity threshold from 9 to 93% based on different *R*_0_ values. However, this range was estimated under experimental conditions [19]. Our study used the actual field data and indicated that at least 80% of pigs in these small-scale farms must be vaccinated to prevent future ASF outbreaks.

We acknowledged that we had only ten farms in the present study. Future research may include a higher number of farms with a broader variety of farm scales to provide a more accurate result. However, the modeling framework constructed in this study is flexible and adjustable to the updated parameters. Our model is thus readily applicable once new data are available. In addition, we identified the new case with symptoms of ASF, including loss of appetite, fatigue, high fever at 40-42 ºC, and lesion on the skin, since it was impossible to detect the infected individuals by PCR due to our financial constraints. Nonetheless, the first cases in all these farms were confirmed with PCR to ensure the ASF-infected status.

## 5. Conclusions

This is the first study that estimates the *R*_0_ values of the 2019 ASF outbreaks in Vietnam. We suggest vaccinating at least 80% of pigs on each farm once a commercially approved ASF vaccine is available. However, this suggestion relied on the results of passive surveillance. Therefore, higher vaccination coverage should be encouraged for achieving herd immunity for cases with laboratory confirmation. The improvement of biosecurity levels in small-scale farms should be seriously considered to prevent and control the spread of ASFV. Culling only infected pigs could reduce the number of infected animals in the herd and lessen the outbreak’s impact.

## Figures and Tables

**Figure 1 vetsci-09-00292-f001:**
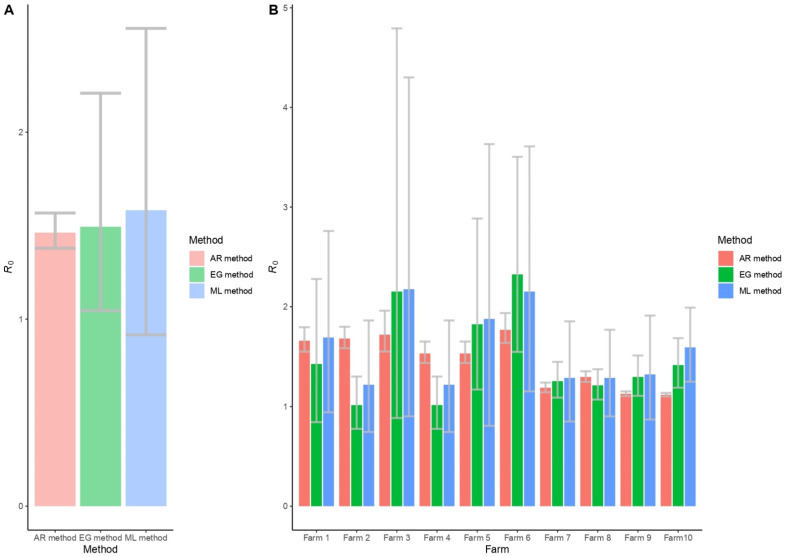
The *R*_0_ values of within-farm ASF transmission are estimated with the exponential growth method (EG), maximum likelihood method (ML), and attack rate method (AR). (**A**) Overall *R*_0_ values summarized from all recruited farms. (**B**) *R*_0_ values of each farm.

**Figure 2 vetsci-09-00292-f002:**
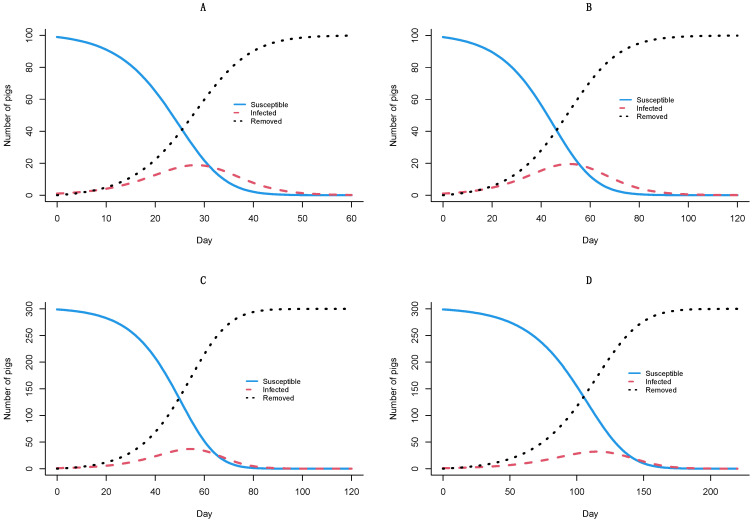
SIR models of ASF transmission within-farm. Models 1 (*T* = 3–6 days) and 2 (*T* = 3–14 days) with the range of infectious period of the farm scale 100 to 299 animals (**A**,**B**) and models 3 (*T* = 3–6 days) and 4 (*T* = 3–14 days) with the range of infectious period of farm scale 300 to 999 animals (**C**,**D**), respectively.

**Table 1 vetsci-09-00292-t001:** Modeling scenarios for the within-farm transmission of ASFV.

Model	Farm Scale (Pig Heads)	Range of Infectious Period (T) (Day)
1	100–299	3–6
2	100–299	3–14
3	300–999	3–6
4	300–999	3–14

**Table 2 vetsci-09-00292-t002:** Model fitting results for a serial interval of each farm.

Farm ID	Mean	Standard Deviation	Best-Fitting Generation Time Distribution
Farm 1	2.790321	2.021366	Gamma
Farm 2	2.166667	1.651193	Gamma
Farm 3	1.916569	1.758193	Lognormal
Farm 4	2.166667	1.651193	Gamma
Farm 5	2.345724	1.59627	Weibull
Farm 6	2.815932	2.068268	Weibull
Farm 7	1.252727	1.270785	Lognormal
Farm 8	2.72771	1.803711	Weibull
Farm 9	2.273011	2.556474	Lognormal
Farm 10	6.703201	2.880502	Weibull

**Table 3 vetsci-09-00292-t003:** Estimated model parameters.

Parameters	Description	Farm Scale			
		100–299		300–999	
*R* _0_	The basic reproduction number	1.66 (95%CI: 0.88–2.84)		1.40 (95%CI: 1.01–1.90)	
*T*	The infectious period (Guinat et al., 2016)	4.5 (Minimum)	8.5 (Maximum)	4.5 (Minimum)	8.5 (Maximum)
*γ*	Removal rate	0.22	0.12	0.22	0.12
*β*	Transmission rate	0.37 (95%CI: 0.20–0.63)	0.20 (95%CI: 0.10–0.33)	0.31 (95%CI: 0.22–0.41)	0.16 (95%CI: 0.12–0.22)

**Table 4 vetsci-09-00292-t004:** Parameter estimation for vaccine coverage requirement.

Parameter	Description	Farm Scale	
		100–299	300–999
E	Vaccine efficacy	80%	
$I_{c}$	Herd immunity threshold	0.3975 (95%CI: 0–0.6479)	0.2857 (95%CI: 0.0099–0.4737)
$V_{c}$	Vaccine coverage	0.4970 (95%CI: 0–0.8099)	0.3571 (95%CI: 0.0124–0.5921)

## Data Availability

Not applicable.

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
