# Peer review of "Dynamic Models of Within-Herd Transmission and Recommendation for Vaccination Coverage Requirement in the Case of African Swine Fever in Vietnam"

_vetsci, 2022, doi:10.3390/vetsci9060292_

Round 1
Reviewer 1 Report
The pig business of Vietnam has been recently affected by the onset of African swine fever (ASF), a highly contagious and fatal disease of domestic pigs, causing large economic loss. This study aimed to develop a disease dynamic model to explain the diffusion of ASF in Vietnamese pig populations and suggest a protective vaccine coverage level required to prevent future outbreaks. The manuscript is well written and organized, but the figures are very difficult to read. I suggest improving the quality of the figures.
Also, the conclusion should be expanded.
Moderate English revision is required.
Author Response
Comment
“The pig business of Vietnam has been recently affected by the onset of African swine fever (ASF), a highly contagious and fatal disease of domestic pigs, causing large economic loss. This study aimed to develop a disease dynamic model to explain the diffusion of ASF in Vietnamese pig populations and suggest a protective vaccine coverage level required to prevent future outbreaks. The manuscript is well written and organized, but the figures are very difficult to read.
Response
We are grateful for these supportive comments.
I suggest improving the quality of the figures.”
Response
We improved the quality of the figures in the MS documents as suggested.
Comment
“Also, the conclusion should be expanded.”
Response
In conclusion, this is the first study that estimates the R0 values of the 2019 ASF outbreaks in Vietnam. We suggest vaccinating at least 80% of pigs on each farm once a commercially approved ASF vaccine is available. The improvement of biosecurity levels in small-scale farms should be seriously considered to prevent and control the spread of ASFV. Culling only infected pigs could reduce the number of infected animals in the herd and lessen the impact of the outbreak.
Comment
“Moderate English revision is required.”
Response
We carefully revised our manuscript again to improve the readability.
Reviewer 2 Report
The Authors professionally approached an interesting qualitative analysis reacting to calulatation of epidemiological paramters for ASF outbreaks in Vietnam. The language and general flow is very communicative which is a great asset to the manuscript (it's really easy to read and understand the authors' ideas.
L 100 Probably should be “total pig amounts”
Data collection subchapter is extremely unclear. Authors once wrote that they use daily “morbidity” once number of deaths. They must tell readers more about outbreak management (what kind of measures were applied when an outbreak was suspected, surveillance system, etcs).
Please add epidemiological curves for these 10 farms to the materials chapter too
Authors wrongly assume that that removal rate is 1/infectious period. \gamma parameters is a combination of natural transmission and measures of infection control. Authors instead allow variability in transmission rates (which is mathematically correct, but is counterintuitive from infection dynamics perspective). Thus, authors must mention in discussion that modification of \gamma could be another possibility to include variability, which mathematically is indistinshed (at least for R0 calculation).
Thus conclusions should also be weakened about critical vaccination coverage (for instance upon what condition it's given as mimal passive surveillance only, etc.)
Author Response
Comment
“The Authors professionally approached an interesting qualitative analysis reacting to calulatation of epidemiological paramters for ASF outbreaks in Vietnam. The language and general flow is very communicative which is a great asset to the manuscript (it's really easy to read and understand the authors' ideas.)”
Response
We are grateful for these supportive comments.
Comment
“L 100 Probably should be “total pig amounts”
Response
This estimation is for total pig farms, not for total pig amounts.
Comment
“Data collection subchapter is extremely unclear. Authors once wrote that they use daily “morbidity” once number of deaths. They must tell readers more about outbreak management (what kind of measures were applied when an outbreak was suspected, surveillance system, etcs).”
Response
Thanks to the reviewer very much for pointing this out. We have corrected as cases, not deaths as follows:
“The number of ASF cases, regardless of whether it was fattening pig or sow, was combined on each farm.”
Comment
“Please add epidemiological curves for these 10 farms to the materials chapter too”
Response
Thank you very much for your comment. In fact, we created the curves and found them difficult to read. We then moved it to Supplementary material.
Comment
“Authors wrongly assume that that removal rate is 1/infectious period. \gamma parameters is a combination of natural transmission and measures of infection control. Authors instead allow variability in transmission rates (which is mathematically correct, but is counterintuitive from infection dynamics perspective). Thus, authors must mention in discussion that modification of \gamma could be another possibility to include variability, which mathematically is indistinshed (at least for R0 calculation).”
Response
We addressed this point as suggested in the discussion part; “Besides, modification of γ could be another possibility to include variability, which mathematically is undistinguished, at least for R0 calculation.”
Comment
“Thus conclusions should also be weakened about critical vaccination coverage (for instance upon what condition it's given as mimal passive surveillance only, etc.)”
Response
We agree with the reviewer and we address this point in the discussion as “We suggest vaccinating at least 80% of pigs on each farm once a commercially approved ASF vaccine is available. However, this suggestion relied on the results of passive surveillance. Therefore, for cases with laboratory confirmation, higher vaccination coverage should be encouraged for achieving herd immunity.”